# Modelling Mechanically Induced Non-Newtonian Flows to Improve the Energy Efficiency of Anaerobic Digesters

**Andrew Oates [1],\*** , **Thomas Neuner [2]**, **Michael Meister [2]**, **Duncan Borman [3]**,
**Miller Camargo-Valero [4,5]** , **Andrew Sleigh [3] and Paul Fischer [2]**

[1]  EPSRC Centre for Doctoral Training in Fluid Dynamics, University of Leeds, Leeds LS2 9JT, UK
[2]  Department of Environmental, Process, and Energy Engineering, Management Center Innsbruck—The Entrepreneurial School, 6020 Innsbruck, Austria; Thomas.Neuner@mci.edu (T.N.); Michael.Meister@mci.edu (M.M.); paul.f1@gmx.net (P.F.)
[3]  School of Civil Engineering, University of Leeds, Leeds LS2 9JT, UK; d.j.borman@leeds.ac.uk (D.B.); P.A.Sleigh@leeds.ac.uk (A.S.)
[4]  BioResource Systems Research Group, School of Civil Engineering, University of Leeds, Leeds LS2 9JT, UK; M.A.Camargo-Valero@leeds.ac.uk
[5]  Departamento de Ingeniería Química, Universidad Nacional de Colombia, Campus La Nubia, 17003 Manizales, Colombia
\*  Correspondence: scajo@leeds.ac.uk

**Abstract:** In this paper, a finite volume based computational fluid dynamics (CFD) model has been developed for investigating the mixing of non-Newtonian flows and operating conditions of an anaerobic digester. A CFD model using the multiple reference frame has been implemented in order to model the mixing in an anaerobic digester. Two different agitator designs have been implemented: a design currently used in a full-scale anaerobic mixing device, SCABA, and an alternative helical ribbon design. Lab-scale experiments have been conducted with these two mixing device designs using a water-glycerol mixture to replicate a slurry with total solids concentration of 7.5%, which have been used to validate the CFD model. The CFD model has then been scaled up in order to replicate a full-scale anaerobic digester under real operating parameters that is mechanically stirred with the SCABA design. The influence of the non-Newtonian behaviour has been investigated and found to be important for the power demand calculation. Furthermore, the other helical mixing device has been implemented at full scale and a case study comparing the two agitators has been performed; assessing the mixing capabilities and power consumption of the two designs. It was found that, for a total solids concentrations of 7.5%, the helical design could produce similar mixing capabilities as the SCABA design at a lower power consumption. Finally, the potential power savings of the more energy efficient helical design has been estimated if implemented across the whole of the United Kingdom (UK)/Austria.

**Keywords:** anaerobic digester; computational fluid dynamics; hydrodynamics; mechanical mixing; non-newtonian flows

## 1. Introduction

Wastewater treatment plants aim to reduce the contaminate level of the water to safe levels, so that the water can be discharged back into the water cycle. Anaerobic Digesters (ADs) are an advanced wastewater treatment technique that reduces the biological residues created during the wastewater treatment cycle while using anaerobic biological processes. During the digestion process, heat and energy, the latter in the form of biogas, is produced and utilized as a renewable energy source [1,2].

A major challenge in anaerobic digestion is efficiency; optimising energy yield whilst reducing the operational costs of the digester. The AD performance is dependent on how easily microorganisms can reach the necessary nutrients under the right environmental settings. The desired conditions for the biological residues inside of digesters are: physical, chemical, and biological uniformity, which can be achieved by thoroughly mixing [3–5]. Additionally, along with improving the yield of biogas production, a thoroughly mixed reactor will also reduce the amount of oversaturation of biogas in the sludge and, thus, minimise the amount of biogas leaving within the effluent.

Various mixing techniques have been implemented across a range of different anaerobic digesters, with each design having different advantages, disadvantages, and energy requirements. The most common mixing methods are: gas mixers, mechanical stirred and mechanically pumped. Among these, mechanically stirred has showed to produce the most efficient mixing when comparing the mixing capabilities to power consumption [6]. The energy consumption of the anaerobic digester is largely dominated by the mixing and, thus, is a continued area of research to optimise the energy balance of the system [7,8].

Full-scale onsite AD measurements and studies are difficult due to biological residues and the process design of the digesters; therefore, lab-scale experiments have been performed to further study different phenomena under controlled conditions [9–11], however, these studies on their own are limited by their large scale difference. An alternative tool for investigating the mixing in anaerobic digesters is computational fluid dynamics (CFD). Hurtado et al. [4] used CFD to model an AD stirred by re-circulation and used the simulation to study residence time distribution, turbulence intensity, and velocity under nominal operating conditions. Coughtrie et al. [12] studied the effect of different multi-phase and turbulence models for a bench scale gas lift digester. Wu [6] analysed the effect of rotating speeds, impeller and draft-tube placement, and total solids (TS) concentration on the mixing in mechanical draft-tube driven cylindrical and egg shaped ADs. Meister et al. [2] used CFD simulations to assess and improve the operation of a mechanical draft tube driven egg-shaped AD. They considered the effect of pumped re-circulation on the generated flow field and concluded design changes in order to improve mixing. Leonzio [8] reviews three different AD configurations using CFD and proposes an innovative mixing system with external recirculating pumps to improve mixing over the traditional designs.

Similar to AD, the mixing of highly viscous fluids is required in a wide range of industrial process, e.g., paint, polymers, food production; and obtaining homogenisation efficiency is challenging [13]. For highly viscous mechanically stirred fluids, inefficiencies can arise from low radii agitators because stagnant zones can form at regions far away from the agitator [14]. To compensate for this, higher rotational speeds are required, but this increases the stirrer power consumption and shear rates in the fluid. In flow where shear sensitivity impacts the reactor performance, the use of low rotational speed agitators is recommended [15], which, in the case of AD, is an important consideration, as high shear rates can impact the biological processes and reduce the biogas production; a desired product of the AD process [16].

Tsui and Hu [14] developed a CFD model to investigate the mixing generated by a helical ribbon blade and the influence of blade pitch on mixing performance. Hosseini et al. [17] also studied the potential of a helical ribbon agitator by manufacturing a batch stirred reactor for bio-diesel production and conducting experiments to assess the designs process performance. Lebranchu et al. [16] investigated the impact of shear stress and impeller design on biogas production through lab-scale experimental and CFD approaches and found that a helical ribbon agitator can increase methane production at lower power demand over other conventional impeller designs. Ameur et al. [13] characterised the mixing and power performance of a range of helical-type agitators and Amiraftabi et al. [15] assessed the performance of a dual helical ribbon agitator for a two-phase stirred tank reactor filled with a shear thinning polymer and investigated the relation between the Reynolds number and power consumption. These studies have shown that the helical ribbon blade is more suitable for stirring fluids with high

viscosity and solid content, due to the designs lower rotational speed and shear rates over normal conventional agitator designs, which is shown to result in improved power performance.

In the first section of this paper we will give a short description of a full-scale AD, including its regular operating parameters, a description of the current agitator design, and an introduction to the proposed helical ribbon design. In the next section, there will be a brief explanation of the experimental setup and methods used to obtain results for validation of the CFD model. The third section will then discuss the complete CFD model, introducing the modelling choices for the non-Newtonian slurry behaviour and a description of how the rotational effects of the agitator are induced, specifically the multiple reference frame (MRF) method. The final section will include: a mesh independence study, an assessment of the MRF method, a comparison of experimental and simulation data to validate the CFD model, and a case study assessing the influence of the non-Newtonian behaviour and comparing the mixing capabilities and power consumption of the two agitator designs at full scale.

## 2. Problem Description

The two identical anaerobic digesters in the Rossau wastewater treatment plant, Innsbruck (Austria), serve as a reference for the modelled geometry and operating conditions for the simulations conducted. The main focus of this paper is to compare a newly proposed agitator design with the current working one to assess its mixing capabilities and power efficiency. The mixing device presently used in the ADs works at an acceptable efficiency for the wastewater treatment plant's desired requirements. Therefore, the current design serves as a baseline standard for the operating conditions, expected mixing parameters, and acceptable power demand. By comparing the simulation results of the current agitator design with the actual operating conditions and with the new proposed device, this can serve as a partial validation of the modelling methodology and provide a means of assessing the operating capabilities of the new agitator design, respectively.

### 2.1. Anaerobic Digester Setup

The ADs are continuously operated in series and have a total volume of $5000\,\text{m}^3$ each, with a fill volume of $4600$–$4800\,\text{m}^3$. The reactors have a maximum diameter of $18.2\,\text{m}$, a maximum fill height of $20.1\,\text{m}$ and a tapered shape at the bottom and top (Figure 1 has an illustration of this). The influent of fresh sludge has an average volumetric flow rate of $375\,\text{m}^3/\text{d}$ that is secreted from the top of the reactor onto the sludge surface. Additionally, a fraction of the effluent is re-circulated back into the reactor at a rate of $400\,\text{m}^3/\text{h}$, where the sludge is extracted from the centre flooring by an outlet pipe that is located along the base of the reactor from a sidewall, giving a total inlet volumetric flow rate of $415.6\,\text{m}^3/\text{h}$. The influent sludge has a TS concentration that ranges from 5.9–9.1%, whereas, inside the reacton, the TS concentrations fluctuate between 2.4–4.3% over a yearly cycle and this sludge is stirred mechanically by an open agitator design, known as SCABA (Figure 1a,b). The dominate focus of this paper is to investigate the performance of these agitator at a TS concentration of 7.5%. The benefit of operating at higher TS concentrations is that there is more biological matter per $\text{m}^3$ of sludge, which increases the rate of biogas production, a desired byproduct of AD. Therefore, if we can demonstrate that the agitators can perform efficiently at these higher TS concentrations, then this, in general, is a better operating condition for the AD.

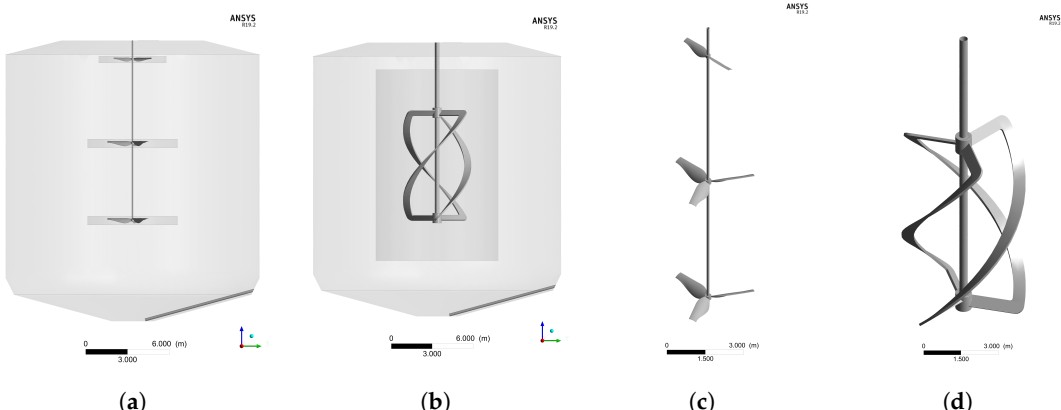

**Figure 1.** Anaerobic Digesters (AD) reactor simulation geometry with the (**a**) SCABA and (**b**) helical ribbon agitators and computer-aided design files of the (**c**) SCABA and (**d**) helical agitators. The cylindrical reactor is based on the Rossau wastewater treatment plant AD, Innsbruck, with radii of 1.5 m, 8.5 m, 9.1 m, 9.1 m and 6 m from the base at respective heights of 0 m, 1.9 m, 3.1 m, 19.1 m, and 20.1 m.

### 2.2. SCABA Mixing Device

SCABA, the name of the mixing device design created by *SULZER*, is a three sectional open axial agitator. The lower two sections have a three hydro-fin blade configuration, equally spaced around the central shaft, which each have a radius of 1.8 m, while the top section has two symmetric hydro-fin blades with identical orientation with a radius of 1.35 m, Figure 1c. The sections from the bottom have heights of 7.16 m, 12.86 m, and 18.68 m from the base, respectively. The agitator is rotated at a speed of 9 rpm and the daily operation of the agitator device is to rotate clockwise for 4 h and then rotate in the opposite direction for 30 min. to stop fouling on the blades which, under continuous operation, consumes 1.5 kW. The hydro-fin blades are all relatively small compared to the radius of the reactor, Figure 1a, which requires them to be rotated at relatively high speeds and, therefore, requires a high power demand. The influence of variations in TS concentrations on the flow profile and power consumption is unknown and, along with a comparison of an alternative agitator design, will be investigated in this report.

### 2.3. Helical Mixing Device

A tri-helical ribbon agitator has been proposed as an alternative mixing device for stirring in a full-scale AD. Previous literature has found that a helical ribbon design can produce the necessary mixing in stirred reactors [11,15,17] at a lower power and shear rates when compared with conventional agitator designs, which are desired characteristics for AD [16]. The proposed helical agitator has a radius of 5 m (approximately 35% of the reactor) and height of 8.14 m, a ribbon blade width of 1 m throughout and was placed at 7 m above the base of the reactor, see Figure 1b,d for illustrations of the helical agitator. The far reaching ribbon design means that a slower rotational speed is required when compared to lower radii agitators, like SCABA, and an appropriate clockwise rotational speed of 2.5 rpm was found to be comparable with the SCABA setup that will be further discussed in the Section 5

## 3. Experimental Setup

Experiments with the two agitator designs were conducted at lab-scale in order to provide velocity data for validation of the CFD modelling approaches. To achieve this, a lab-scale acrylic cylindrical glass reactor with a tapered base (replicating the full-scale design in Figure 1a,b), a linear scaled inner diameter of 0.22 m, and fill volume of 7.948 L was manufactured and investigated. Velocity data were obtained by using a GE Logiq 300 ultra sonic device that emits ultrasound waves through

the reactor and detects variations in the signal as it transitions through density changes, which is converted to a real-time video. The analog video output is then converted to digital frames with a frame grabber device. Ultrasonic gel was used between the measuring head and reactor walls to prevent signal interference from the transition medium. Clockwise mixing was induced by a Heidolph RZ 2102 electronic stirrer with variable speed control. An IKA R3003.1 helical mixing device and a three-dimensional (3D)-printed scaled version of the existing SCABA device (Figure 1) were used for the investigation of the flow profile. The SCABA mixing device was scaled using the constant specific power in accordance to the Penney-approach for mixing of high viscous fluids [18,19]. The dimensions as well as the operating parameters of the lab-scale mixing devices can be found in Table 1.

**Table 1.** Dimensions of the lab-scale mixing devices used for mechanical agitation in the experiments.

| Agitator Design | Diameter (m) | Height (m) |
|---|---|---|
| Helical | 0.1 | 0.1628 |
| SCABA | 0.12/0.12/0.09 | 0.152 |

In order to simulate the rheological properties of the sludge wastewater, a mixture of water and glycerol was used as a Newtonian fluid substitute [20]. Specific water and glycerol mixing ratios have viscosities and densities that can represent different TS concentration ranges; these ratios can be found in Table 2.

**Table 2.** Fluid properties of the glycerol/water mixtures used in the experiments.

| TS (%) | Glycerol/Water | $\eta$ (Pa s) | $\rho$ (kgm$^{-3}$) |
|---|---|---|---|
| - | 0/100 wt% | $9.8 \times 10^{-4}$ | 997.83 |
| 2.5 | 50/50 wt% | $5.8 \times 10^{-3}$ | 1126.3 |
| 5.4 | 60/40 wt% | $1.062 \times 10^{-2}$ | 1153.5 |
| 7.5 | 75/25 wt% | $3.43 \times 10^{-2}$ | 1193.7 |
| 9.1 | 85/15 wt% | 0.11368 | 1222.7 |
| >12 | 100/0 wt% | 1.2901 | 1260.1 |

*Experimental Procedure*

Polyamide tracer particles were added to the fluid in order to improve data quality by increasing the number of density transitions which are detected by the ultrasonic device. The nearly massless particles had a mean diameter of 0.005 μm with almost zero buoyancy, such that they travelled with the fluid flow. The changes in the ultrasound waves that were generated by these particles can then be solved and tracked homogeneously in the fluid to generate a velocity vector field. The gathered ultrasonic frames were evaluated by cross-correlation using PIVLab with a FFT window deformation algorithm with incrementing sequencing style [21]. Every radial and tangential particle shift is computed for at least 1000 frames; shifts along the height or vertical directions are not considered with this measuring approach. The measuring process is conducted in increments of 0.01 m along the reactor height starting at 0.03 m. An example of the PIVLab evaluation is illustrated in Figure 2 for the helical device with a 50/50 wt% glycerol-water mixture at a reactor height of 0.03 m; the region of interest is the whole two-dimensional (2D) sliced flow field. The absolute average velocity is computed using the single translocation of the particles as well as the time interval between the single frames. Therefore, the ultrasonic measurement is restricted in terms of flow velocity due to a sampling rate of 24 Hz and associated aliasing effects at higher rotational speeds. The time averaged velocity values were computed over all measured frames to be compared with CFD data and, therefore, measurement intervals were set to ensure that at least a minimum of 1000 frames were considered per experimental setup. For validation, CFD simulations representing the same experimental setup were conducted. In the lab-scale CFD simulations, Newtonian behaviour was set with constant viscosity in order to better represent the glycerol-water mixtures used in the experimental setup.

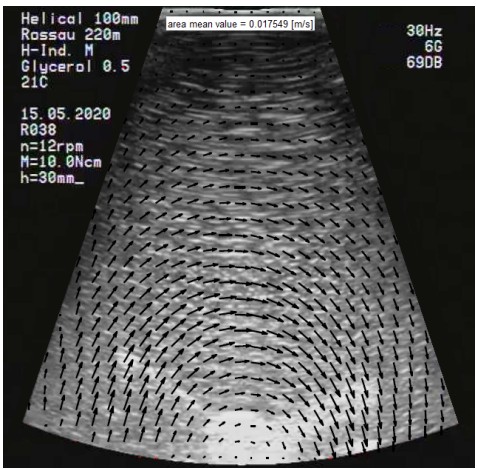

**Figure 2.** Illustration of captured and evaluated flow field in the X-Y-plane using the ultrasound method. A 50/50 wt% glycerol-water mixture was used with the helical stirring device rotating at 12 rpm ($h$ = 0.03 m, $T$ = 21 °C, $\eta$ = 0.00098 Pa s , and $\rho$ = 997.83 kg m$^{-3}$) .

## 4. Methods

The CFD simulations were conducted while using the CFD software ANSYS Fluent version 19.2, geometries and mesh were generated using ANSYS Design Modeler and Meshing respectively [22]. The generated mesh was made with unstructured tetrahedral cells and solved while using the Finite Volume Method (FVM), which is appropriate for its support of unstructured meshes and conservation of physical properties [23]. In these CFD models, the hydrodynamics generated by each agitator was numerically simulated.

*4.1. Governing Equations*

4.1.1. Continuity and Momentum Equation

The steady state Navier–Stokes equations were solved for an isothermal and incompressible fluid. Thus, the continuity equation simplifies to:

$$\nabla \cdot \boldsymbol{u} = 0, \tag{1}$$

where $\boldsymbol{u}$ is the velocity field in the inertial frame or the absolute velocity, and ensures the velocity field is divergence free everywhere. Additionally, the momentum equation, in terms of the absolute velocity, simplifies to:

$$\nabla \cdot (\boldsymbol{uu}) = -\frac{\nabla p}{\rho} + \nabla \cdot \bar{\boldsymbol{\tau}} + \frac{\boldsymbol{F}}{\rho}, \tag{2}$$

where $p$, $\rho$, $\bar{\boldsymbol{\tau}}$ and $\boldsymbol{F}$ are the fluid's pressure, density, viscous stress tensor, and the body forces, respectively, and the viscous stress tensor, $\bar{\boldsymbol{\tau}}$, is given by:

$$\bar{\boldsymbol{\tau}} = \frac{\mu}{\rho}(\nabla \boldsymbol{u} + \nabla \boldsymbol{u}^T), \tag{3}$$

where $\mu$ is the local dynamic viscosity of the fluid.

4.1.2. Multiple Reference Frame

To generate the flow field, the effect of induced mixing by the rotating agitator needs to be modelled. This can be achieved while using the MRF method, which, for the size of full-scale ADs, is a computational cost appropriate method and is a regularly adopted and recommended approach to

modelling mixing in AD [2,3,11,13]. In this method, a cylindrical zone surrounding the mixing device is created in the domain and a change of reference frame is made in this new region, the rotating reference frame (RRF). In the RRF, the agitator is now stationary and the fluid surrounding rotates at a relative velocity, and continuity is enforced at the boundary between frames to have a smooth transition [24]. Under constant rotational velocity, we can relate the absolute and relative velocity by:

$$\boldsymbol{u} = \boldsymbol{u_r} + (\boldsymbol{\Omega} \times \boldsymbol{r})$$

where $\boldsymbol{u_r}$ is the relative velocity and $\boldsymbol{r}$ is the position vector in the RRF. In the RRF, the momentum equations defined in Equation (2) are supplemented by additional acceleration terms that appear when accounting for the time dependent rotation of the axes. Therefore, the steady state momentum equations in the RRF in terms of the absolute velocity becomes:

$$\nabla \cdot (\boldsymbol{u_r u}) + \boldsymbol{\Omega} \times \boldsymbol{u} = -\frac{\nabla p}{\rho} + \nabla \cdot \bar{\tau} + \frac{\boldsymbol{F}}{\rho}, \tag{4}$$

where the $\boldsymbol{\Omega} \times \boldsymbol{u}$ term captures the additional rotational effects. The RRF needs to capture the direct rotational effects that are generated from the mixing device and it is recommended that the aspect ratio of the frame to agitator dimensions exceeds 1.5 for both diameter and height [3], such that no unphysical behaviour arises between the two frames. A further investigation of the aspect ratio size will be made in the Section 5

### 4.1.3. Turbulence Closure

The Reynolds number, *Re*, for mechanically stirred non-Newtonian fluids can be defined as:

$$Re = \frac{\rho N d_a}{\eta}, \tag{5}$$

where $\rho$ and $\eta$ are the slurry density and viscosity, respectively, $N$ is the rotational speed and $d_a$ is the agitator diameter [3]. The calculated *Re* for the flows simulated in this report implies that they all lie within turbulent regime and, therefore, accurately capturing the turbulence effects is important for correctly predicting the hydrodynamics. The Reynolds-Averaged Navier–Stokes (RANS) formulation has been used in the simulations to solve for the flow field that requires additional modelling of the Reynolds stresses for closure. Various turbulence models exists and their ability to reproduce the agitator induced mixing in AD have been extensively investigated in Wu [3] and Meister et al. [2], and both recommend the realizable $k - \epsilon$ and $k - \omega$ for predicting the mechanical agitation for non-Newtonian fluids in AD. In this paper, the two-equation realizable $k - \epsilon$ model has been adopted for the turbulence closure in the CFD simulations. The focus of this paper is not a thorough comparison of the various turbulence models, however, a study of the realizable $k - \epsilon$, standard $k - \omega$, and SST $k - \omega$ models was conducted to analyse the differences in the model choices. The formulation of the realizable $k - \epsilon$ contains extra mathematical constraints that improves its robustness and has been shown to have substantial improvements for flows with strong rotation over the standard $k - \epsilon$ model [22]; the formulation derivation can be found in Shih et al. [25] and the equations and constants used in the equations can be found in ANSYS-Fluent theory [22].

### 4.1.4. Near Wall Treatment

In coarse grids where the resolution does not reach the viscosity-affected inner regions, wall functions are required to model the fluid behaviour near walls by exploiting the universal logarithmic wall law; most previous papers use either standard wall functions [2,4,10] or do not mention them. If the bulk mixing in the middle of the domain is of interest, rather than the wall forces, then wall functions are acceptable. However, if wall forces are important for the generation of the flow, wall functions accuracy deteriorates as refinement is made near the wall where the resolution

lies inside of the viscosity-effected region. In open agitator stirred reactors, the blades drive the flow and, therefore, it is important to accurately capture these wall features in order to correctly predict the fluid flow. The initial simulations showed that not correctly modelling the wall behaviour can impact fluid flow and power calculations; illustrating the need to accurately model the wall behaviour. Thus, inflation layers in the grids were used to increase the resolutions around the walls of the whole geometry, which resulted in the near wall cell lying inside of the viscosity-affected regions. At these resolutions, the $y^+$ values (non-dimensional normal distances from the wall) are such that wall functions become inaccurate and alternative and higher order wall modelling needs to be considered [12,22]. Therefore, enhanced wall treatment has been used for all CFD simulations in order to accurately resolve the near wall features. This does require additional computational power, however, it was found that the enhanced wall treatment did not substantially increase simulation run time.

### 4.2. Non-Newtonian Modelling

The properties of sludge are dictated by the compositions of the dry matter content that is determined by the origins of the influent wastewater. The most common feature is the non-Newtonian behaviour of sludge that acts as a shear-thinning fluid [26]. An common approach to characterise the slurry manure rheological properties is describing the TS concentration; describing the percentage of the slurry that is solid matter. Experiments have been conducted in order to measure the non-Newtonian fluid properties of slurries at different temperatures for a range of TS concentrations and to fit non-Newtonian models to them [27–29]. Understanding and modelling the sludge rheological properties in simulations is important for accurately predicting the flow behaviour of the stirring by the agitators [30]. The most common method to replicate the non-Newtonian behaviour of slurry in CFD simulations is to approximate the fluid as single phase with constant density and use the power law model that has been matched to experimental data in [26–28] and been adopted into AD CFD studies in [1–3,8,16], while the alternative Herschel-Bulkley and Bingham model has been fitted to experimental data from [29,30] and applied to CFD simulations in [4,31].

Power Law Model

The power law for non-Newtonian fluids calculates the dynamic viscosity, $\eta$, from the shear rate $\dot{\gamma}$, according to, under isothermal conditions:

$$\eta = k\dot{\gamma}^{n-1},$$ (6)

where $k$ and $n$ are the consistency and power-law index respectively. Values of $n < 1$ describes a non-Newtonian shear-thinning fluid that are the properties generally seen in sludge [26,30]. An alternative form of this model is the capped power law model where additional limits are placed on the dynamic viscosity, $\eta_0$ and $\eta_\infty$, in order to restrict the values of the dynamic viscosity to correspond with experimentally measured viscosity values from specific explored shear rate ranges [22], which for a shear-thinning fluid is defined as:

$$\eta = \begin{cases} \eta_0 & \text{for } \eta \geq \eta_0, \\ k\dot{\gamma}^{n-1} & \text{for } \eta_\infty \leq \eta \leq \eta_0, \\ \eta_\infty & \text{for } \eta \leq \eta_\infty. \end{cases}$$ (7)

The capped power-law model has been fitted to experimental slurry data by Achkari-Begdouri and Goodrich [28], such that, at constant temperature 35 °C, the non-Newtonian sludge behaviour for varying TS concentrations can be summarised by the values in Table 3.

**Table 3.** Sludge wastewater properties and coefficients for the non-Newtonian simple power law model at different total solids (TS) concentrations [28]; where $\dot{\gamma}_0$–$\dot{\gamma}_\infty$ describes the range of shear rates that are valid in the power law for each TS concentration, such that the calculated viscosity values are inside $\eta_0$–$\eta_\infty$.

| TS (%) | $\rho$ (kg m$^{-3}$) | $\eta_0$–$\eta_\infty$ (Pa s) | k (Pa s$^n$) | n | $\dot{\gamma}_0$–$\dot{\gamma}_\infty$ (s$^{-1}$) |
|---|---|---|---|---|---|
| 2.5 | 1000.36 | $6 - 8 \times 10^{-3}$ | 0.042 | 0.710 | $702 - 226$ |
| 5.4 | 1000.78 | $1 - 3 \times 10^{-2}$ | 0.192 | 0.562 | $702 - 50$ |
| 7.5 | 1001.00 | $0.3 - 1.7 \times 10^{-1}$ | 0.525 | 0.533 | $399 - 11$ |
| 9.1 | 1001.31 | $0.7 - 2.9 \times 10^{-1}$ | 1.052 | 0.467 | $156 - 11$ |
| 12.1 | 1001.73 | $0.25 - 2.93$ | 5.885 | 0.367 | $149 - 3$ |

Whilst the power law models are the simplest non-Newtonian models, they have been adopted in multiple AD CFD studies. Lebranchu et al. [16] used the power law model (Equation (6)) in their investigation of biogas production in AD, while [1–3,8] have all adopted the capped power law model utilising the extensive experimental data set for a range of TS concentrations, Table 3, provided by [28]. The capped power law model is adopted in the majority of this work in order to model the non-Newtonian behaviour of sludge at TS concentrations of 2.5–12.1% using the sludge property data in Table 3 due to the extensive use in literature and our interest in the hydrodynamic impact of varying TS concentrations. In summary, the sludge in the reactor will be modelled as single-phase, incompressible, isothermal at constant temperature of 35 °C, and non-Newtonian described by the capped power law.

### 4.3. Simulation Initialization

#### 4.3.1. Solution Method and Initial Conditions

The SIMPLE scheme is used to couple pressure and velocity. The least square based discretization approach is specified for the gradients. The second order scheme and upwind approach are used for the pressure and momentum, respectively, and the first order approximation is specified for the turbulent kinetic energy and dissipation rate. The full multigrid initialization method is used to decrease convergence time. The realizable $k - \epsilon$ model is used for turbulence closure. The default under-relaxation factors were used until convergence, then reduced by a factor of 6 and ran until convergence was reached again [22]. The residuals, power, velocity probes, and outflow were all monitored in order to confirm solution convergence.

#### 4.3.2. Boundaries and Cell Zone Conditions

No-slip boundaries with a default roughness constant of 0.5 were specified for all walls [22]. A volume flow rate of 0.1155 m$^3$/s was defined at the inlet and the outlet pipe was defined as a zero gauge pressure outlet. The inlet and outlet have a diameter of 1.5 m and 0.2 m, respectively. The sludge TS concentration for each case was defined the same for both the sludge entering and inside the reactor. For the full-scale simulations, the MRF aspect ratio for the width/height was defined as 1.8/1.8 and 1.8/1.72, with a rotational velocity of 9 rpm and 2.5 rpm, for the SCABA and helical agitator, respectively.

## 5. Results and Discussion

### 5.1. Mesh Independence Study

A mesh independence study has been conducted while using a tetrahedral unstructured mesh, generated in ANSYS meshing software (version 19.2), for the lab and full scale simulations for both agitators. An inflation layer was applied along the mesh surfaces to resolve near wall hydrodynamic features, see Figure 3. Table 4 presents the mesh details.

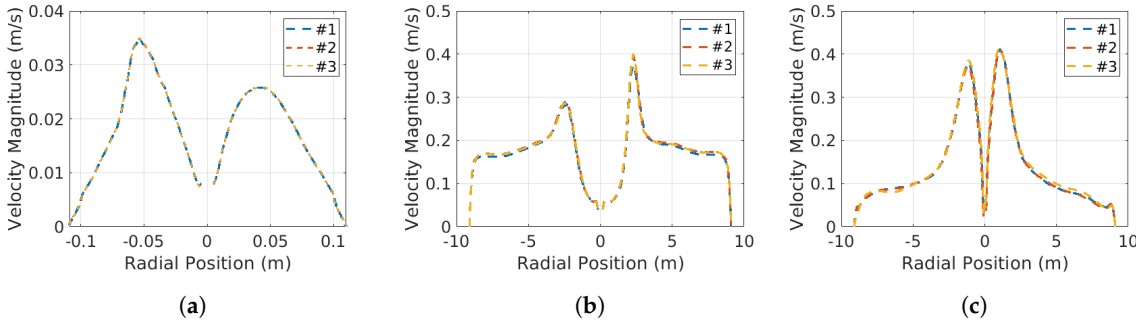

**Figure 3.** Velocity magnitude profile along the AD reactor diameter at a height of 0.12 m for the lab-scale with the SCABA agitator (**a**) and at a height of 10 m for the helical (**b**) and SCABA (**c**) agitators; comparing the three finest grids.

**Table 4.** Mesh details for the lab and full scale simulations.

| Mesh Number | Lab-Scale Cell Number (mil) | Full-Scale Helical Cell Number (mil) | Full-Scale SCABA Cell Number (mil) |
|---|---|---|---|
| # 1 | 2.7 | 4.5 | 3.7 |
| # 2 | 3.1 | 6.3 | 4.3 |
| # 3 | 3.4 | 8.3 | 5.4 |

Velocity profiles along the reactor diameter at half height are illustrated for the different mesh resolutions presented in Figure 3, showing mesh independence for the two finest meshes. Further mesh independence studies not shown in this report were conducted, which confirmed the results that were found in Figure 3. Therefore, mesh number #2 has been adopted for the lab and full scale agitator simulations.

### 5.2. MRF Study

In these simulations, the MRF has been used to replicate the rotational effects of the stirring agitator due to its balance between the accuracy and computational cost for full-scale AD reactor simulations. Multiple papers have adopted this method for modelling mixing in cylindrical tanks [2,3,11,13,32–34]. However, many of these papers just state their aspect ratio choice without any explanation [11,13] or do not even state the aspect ratio value at all [32–34]. Wu [3] recommends that the aspect ratio value exceeds 1.5 for both height and diameter, so that no unphysical behaviour occurs at the boundary between frames, which may arise due to the enforced continuity at the boundary [24]. However, Wu [3] does not discuss the implications of different aspect ratio values on the predicted hydrodynamics.

In this MRF study, two full-scale helical TS 7.5% simulations were run with aspect ratios of 1.8/1.72 and 1.5/1.5 (these represent diameter/height values of 9 m/14 m and 7.5 m/12.2 m, respectively) in order to compare and understand the impact of varying RRF size on the simulation results and to provide additional guidance for using the MRF in future studies. The plots will not be illustrated here as it is not a focus of this paper, however, the results found that the 1.8 and 1.5 aspect ratio simulations produced almost identical flow fields and velocity profiles. There are slight differences between the results, which implies the aspect ratio can impact the simulations but, for the key features we are interested in: velocity vector field, velocity contributions and power demand; they are practically identical for both simulations. However, it was found for simulations with aspect ratios much larger, order of 2.5, and smaller, order 1.1, the size does begin to significantly effect the predicted flow field. At much smaller aspect ratios, below the range that was recommended by Wu [3], the direct influence of the rotating agitators reaches past the frame dimensions and, therefore, the enforced continuity at the boundary starts to create non-physical behaviour. If the frame is too large it can begin to impede on the vessel walls and, in a similar way, the near wall hydrodynamic features also begins to create non-physical behaviour when the continuity is enforced at the boundary.

What is important to consider when choosing an aspect ratio is the geometry and size of your agitator in comparison to the surrounding vessel geometry. Wu [3] and Meister et al. [2] both had draft tube based stirred reactors and so their MRF aspect ratio diameter was limited by the draft tube, using values less than 1.5. In their case, this was ideal, as it is expected that the fluid is directly influenced by the rotation of the impeller right up to the confined walls of the draft tube. In an open mechanically stirred AD reactors, the reactor walls are the limiting factor for choosing a MRF aspect ratio and their dimensions are almost always significantly (3–6 times) larger than the agitator dimensions. Therefore from this study, it is suggested that a MRF aspect ratios between 1.5 and 2.0 should be used for capturing the influence directly created from the stirring agitator in an open mechanically stirred AD reactor. These ratio values imply that the direct influence of the stirring agitator reaches up to 150–200% of the agitator dimensions in the radial and vertical directions. Choosing values in this range should give very similar results for the bulk hydrodynamics and limited difference in more sensitive parameters without the loss of generality.

### 5.3. Turbulence Modelling Study

The present paper does not aim to conduct a thorough investigation of the turbulence model; however, a turbulence model study is conducted to assess the difference between using the realizable $k - \epsilon$, standard $k - \omega$ and SST $k - \omega$ models. The standard and SST $k - \omega$ models generated very similar flow fields, while the realizable $k - \epsilon$ predicted an overall faster flow field. The realizable $k - \epsilon$ predicts a higher amount of turbulence kinetic energy in the reactor, which also spreads further before being dissipated as compared with the other two models. This difference implies that the realizable model that predicts more momentum is transferred due to turbulence eddies compared with the other two models, such that there is an increased rate of turbulence mixing. This increased rate of turbulence mixing could explain why the flow field is overall faster in the realizable case, as the momentum is being transferred out into the reactor by turbulent eddies. The realizable $k - \epsilon$ also calculated a slightly higher power demand than the other two models. Furthermore, the realizable $k - \epsilon$ was found to be computationally cheaper than the other models and converged faster. Wu [3] investigated various turbulence models for mechanical agitation in AD and found that the realizable $k - \epsilon$ model produced the lowest errors at TS concentrations 5.4%, 9.1%, and 12.1%, performed better than the other non $k - \epsilon$ models at a TS concentration 7.5% and, if not considering the more complex Reynolds stress turbulence model, recommended using the realizable $k - \epsilon$ model. Meister et al. [2] also recommended the realizable $k - \epsilon$ model in their turbulence model study for mixing AD. Therefore, the realizable $k - \epsilon$ model was chosen for modelling the turbulence effects from rotational agitation and it has been adopted to close the RANS equations for the simulations of this paper.

### 5.4. Experimental Results and Validation

The average and weighted average plane velocities at every 0.01 m height increment was computed for the experiments and CFD simulations, respectively, for both agitator devices at two different glycerol-water mixtures, these results are illustrated in Figures 4 and 5. The rotational speed was set at 12 rpm, which was derived from the scaling of the 9 rpm full scale reactor using the Penney-approach, as described in Section 3 [18,19]. The agitator was rotated for 4 min before taking measurements to allow the fluid flow to reach a steady state. Radial and tangential velocity components along a plane at a specific height were recorded and time-averaged over a 45 s recording period. The plane of temporally averaged velocities were then spatial averaged over the whole plane to give single time and spatial averaged 2D planar velocity for a given height. This was repeated five times to produce five independent values. Finally, the mean and standard deviations of these five values were calculated. This was repeated for every 0.01 m height interval considered and the results and standard deviations for each height is presented in Figures 4 and 5. As illustrated in Figures 4 and 5, the CFD flow profiles match well with the experimental results for both of the fluids and agitator designs. The SCABA simulations correctly predict the higher velocities near the blades

at height intervals 0.08 and 0.15 m and capture the general profile that is seen in the experiments. The SCABA simulation for the 75/25 wt% fluid slightly underestimates the velocities near the top of the tank, which could be due to the modelling choice of not including the fluid surface, which can have an impact on the velocities near the top of tank; however, this is not seen in the other simulations. The helical experimental plots in Figure 5 both show a uniform profile throughout the majority of the reactor which is expected near the helical blades due to the symmetrical design and is captured in the helical simulations. The helical simulations do underestimate the velocities near the base of the reactor away from the helical blade for both fluids. This may be due to the simulations not correctly capturing the effects of the tapered base design on the flow field for the helical agitator. Overall, the simulations match very well for both fluids and agitator designs, thus illustrating the validity of the CFD modelling choices and, therefore, these choices have been adopted in the full-scale simulations.

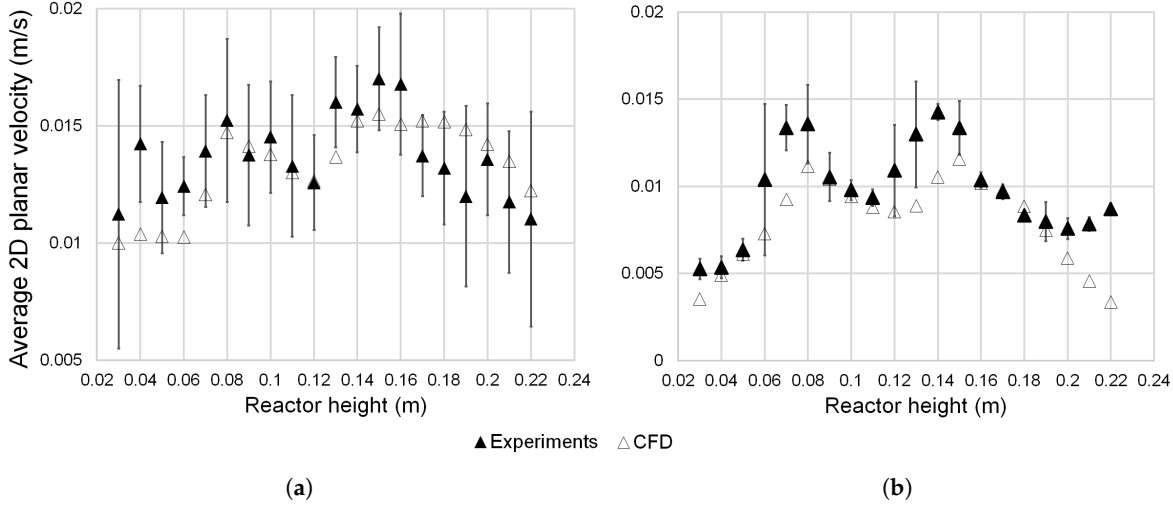

(**a**)              (**b**)

**Figure 4.** Average 2D plane velocities of the experimental data and computational fluid dynamics (CFD) simulations for a rotational speed of 12 rpm and at height intervals of 0.01 m for the SCABA device with pure water (**a**) and a 75/25 wt% glycerol-water mixture (**b**).

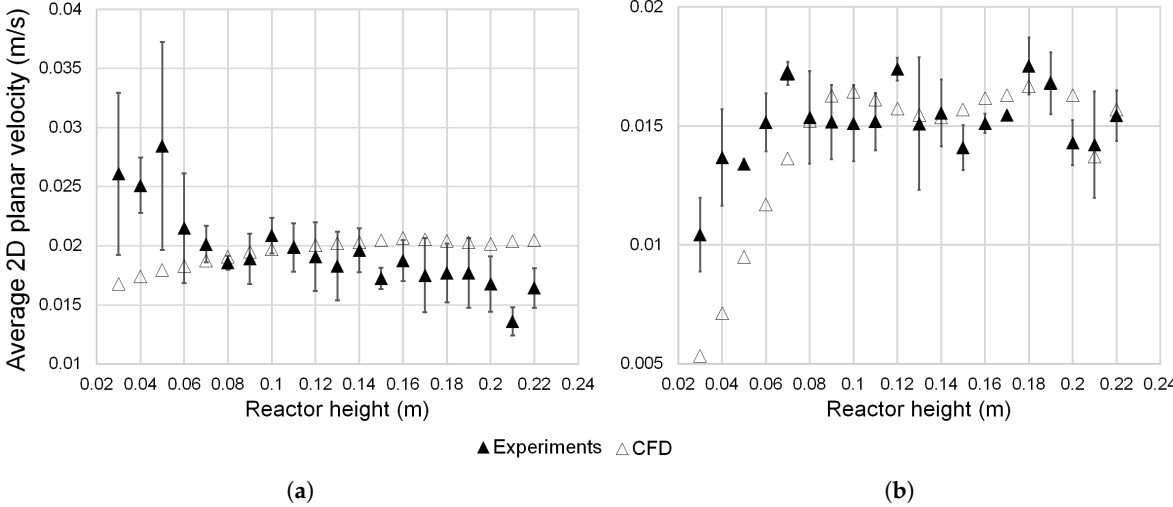

(**a**)              (**b**)

**Figure 5.** Average two-dimensional (2D) plane velocities of the experimental data and CFD simulations for a rotational speed of 12 rpm and at height intervals of 0.01 m for the helical mixing device with pure water (**a**) and a 75/25 wt% glycerol-water mixture (**b**).

*5.5. Full-Scale Case Study: Comparison of SCABA and Helical Mixing Devices*

In the case study, simulations with varying TS concentrations are conducted for the helical and SCABA agitators in order to assess and compare the mixing capabilities and power demand

of the two designs. The SCABA simulations are based on the operating conditions of the Rossau reactor, as described in Section 2. The helical agitator has a larger radii design and applying the same rotational operating condition as the Rossau reactor, 9 rpm, creates an overly-mixed reactor that is both undesirable and not comparable to the SCABA simulations. A previous study has been conducted and found that an operating rotational speed of 2.5 rpm for the helical device creates similar conditions that are comparable with the SCABA simulations. Therefore, the helical simulations were run with identical operating conditions as the Rossau SCABA simulations with the exception of the rotational speed being reduced to 2.5 rpm. The capped non-Newtonian power-law model using parameters from Table 3 is adopted for each TS concentration and the realizable $k - \epsilon$ model was used to close the RANS equations. The other initial conditions have been described in Section 2, above.

### 5.5.1. Influence of the Non-Newtonian Behaviour

The capped power-law model is the most commonly adopted approach in AD CFD papers [1–3,8] and, therefore, has been adopted to model the non-Newtonian behaviour of AD sludge. However, an issue of the model is that, for shear-rates outside of the defined range, $\dot{\gamma}_\infty - \dot{\gamma}_0$, the model assumes constant viscosity and the non-Newtonian behaviour is not modelled. For the simulations that were conducted in these studies, it has been found that the shear-rates lie below $\dot{\gamma}_0$ and, therefore, most of the reactor is modelled with constant viscosity for all TS concentrations. The only locations where the shear-rates are large enough to change the viscosity where at the outlet and near the agitator blades, as can be seen in Figure 6. It was found that the size of the region near the agitator blades influenced by the non-Newtonian model grew larger as the TS concentration increased, which is due to a decrease of the lower limit shear-rate value, $\dot{\gamma}_0$, as higher TS concentrations. On initial inspection, the influence of the non-Newtonian behaviour seems to be minimal. However, the small difference in viscosity near the blades was found to have an impact on the mixing potential of the generated flow field and a reduction in the the power demand calculation at higher TS concentrations. The rotating blades are the dominate source of momentum in the fluid as they drive the flow and, therefore, changes in the fluid properties near the blades can further impact the rest of the hydrodynamics in the reactor. Furthermore, the power demand is calculated from the forces near the agitator walls, where the viscosity values are lower due the non-Newtonian behaviour. This results in the fluid being easier to move in the region near the blades and, therefore, requires less power to stir. This study shows that, even small regions of non-Newtonian behaviour can significantly influence the predicted hydrodynamics for higher TS concentrations and, therefore, the capped power law model has been adopted to capture these effects for both the higher and lower TS concentration simulations to ensure consistency.

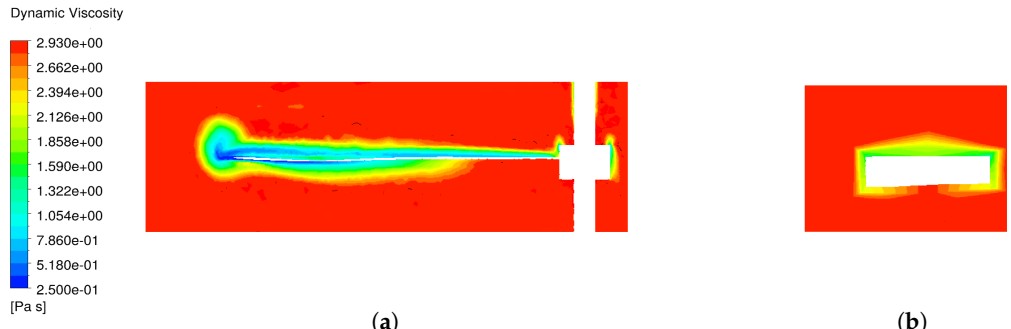

**Figure 6.** Viscosity profiles near the blades of the SCABA (**a**) and helical (**b**) agitators at a TS concentration of 12.1% using the capped power-law model.

### 5.5.2. Flow Profile Analysis

In this section, we will compare the generated flow fields from the two agitator designs for the TS 7.5% fluid. Considering the velocity magnitude plots in Figure 7, there is significant difference between the velocity profiles of the two agitators. The velocity magnitudes in outer regions are larger

for the helical agitator while the velocity magnitudes near agitator blades are significantly larger for the SCABA mixing device. The larger velocities that are near the SCABA blade are due to the higher rotational speed of the blades. At 9 rpm and for hydro-fins with radii of 1.5 and 1.75 m, the outer edges of the SCABA blades travel at 1.41 and 1.65 m/s, respectively. In comparison, the helical agitator rotates at 2.5 rpm with radii between 1.5 and 2.5 m for the main blade which results in the outer blade faces travelling at 0.4 and 0.65 m/s respectively. Approximately 54% of the SCABA blade will travel faster than the helical agitator and that explains why the fluid moves significantly faster near the blades in the SCABA simulations. However, the blades of the helical device have an overall larger surface area that reaches further out into the reactor when compared with the SCABA agitator and, therefore, the fluid being driven by the helical device does not have to travel as far from the agitator to reach the other regions of the reactor. This results in the fluid, on average, moving faster in the helical reactor when compared with the SCABA device, even though the agitator is rotating slower.

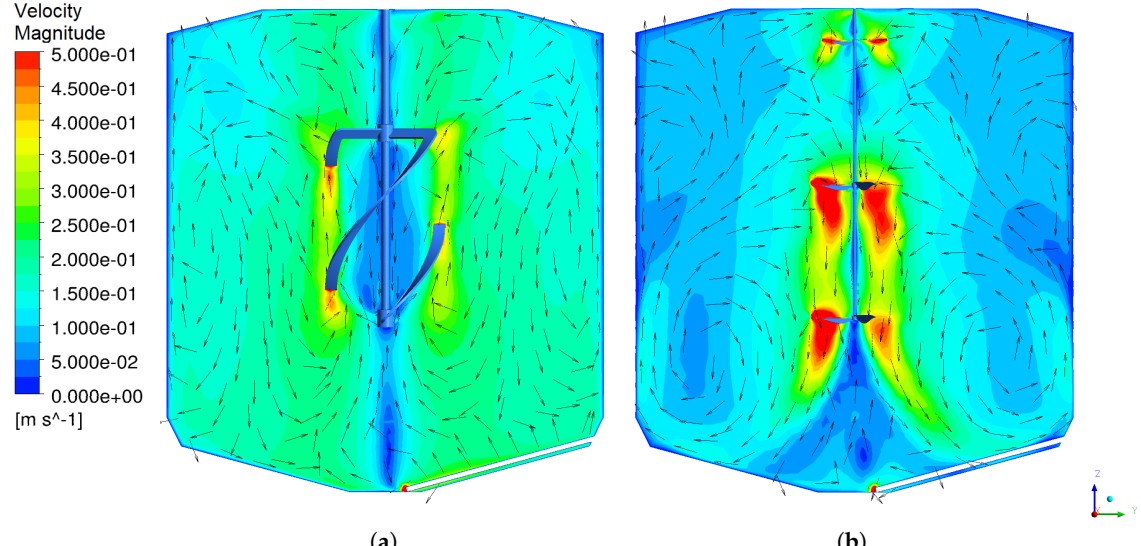

(**a**)  (**b**)

**Figure 7.** Contour and normalised vector plots of the velocity magnitude for the helical (**a**) and SCABA (**b**) agitators at a TS concentration of 7.5%.

In anaerobic digestion, you want uniformity for the biological processes in the reactor and, so, having overall higher average velocities is more important for the process than small regions of very fast moving fluid. Calculating the averaged weighted velocity magnitude for the two agitator simulations above, we find that the SCABA has an average weighted velocity magnitude of 0.104 m/s, while the helical agitator has a value of 0.167 m/s. The higher average velocity for the helical agitator is a more desired AD condition and further emphasises that the far reaching helical design results in an overall increase in the fluid velocity in the reactor. This is also found to be true for all TS concentrations, as seen in Table 5.

**Table 5.** Weighted average velocity magnitudes for the two agitators at different TS concentrations.

|  |  | TS Concentration (%) | | | | |
|---|---|---|---|---|---|---|
|  |  | 2.5 | 5.4 | 7.5 | 9.1 | 12.1 |
| Weighted average velocity magnitude (m/s) | SCABA | 0.113 | 0.117 | 0.104 | 0.100 | 0.0721 |
|  | Helical | 0.212 | 0.188 | 0.167 | 0.160 | 0.109 |

If we consider the vortices presented in Figure 7, we observe three symmetrical vortices for the helical device, while only two are observed in the SCABA case. Both agitators rotate clockwise, however, from Figure 8, we observe that the SCABA device drives the fluid downwards while the

helical device pushes it upwards. This difference is caused by the agitator blade orientation with respect to the rotational directions that are opposite one another for the two agitators. Drawing fluid upwards would seem the desirable agitator operation as long hydraulic retention times are required for AD and, with the outlet near the base of the reactor, it would be expected that the optimal flow field would aim to keep the fluid away from the outlet for as long as possible. The Rossau SCABA operation does not produce such a flow field, however, another desirable feature from the flow field is to have high enough shearing near the base to stop settling of the sludge. Therefore, the SCABA design might be oriented in this way in order to generate the required shearing near the base. Another motive for this orientation is that the clockwise rotation is less power intensive than the alternative, which is more ideal for the AD operation.

Assessing the horizontal or cross velocities, as in Figure 9, we can see that the cross velocities are much faster in the helical device compared with the SCABA device. The slower SCABA velocities may be due to the blade orientation favouring driving the flow in the vertical direction while the helical device drives the flow in a more horizontal direction. This is evident in Figure 8 by the large regions of fluid near the SCABA blade with high z velocities when compared with the rest of the reactor and suggests that the SCABA design is more similar to a draft tube impeller mixer design [2]. Additionally, we observe asymmetrical contours in Figure 9b for the SCABA device, which is due to the asymmetrical three blade design of the agitator. Finally, an issue with the helical design is the large static zone in the centre of the agitator, as illustrated in Figures 7a and 9a, and is a flaw with the far reaching blade design. This could be solved by having an additional smaller helical blade at the centre that could be orientated in the opposite direction, which would create additional shear and turbulence in the central region in order to improve mixing. We can observe from these velocity profiles that the two agitator designs create significantly differently flow fields. The helical device is observed to create an overall faster flow field over the SCABA agitator due to its far reaching design that produces a more homogeneous flow field with smaller velocity gradients throughout the reactor, which are both important flow conditions for the biological processes in the AD.

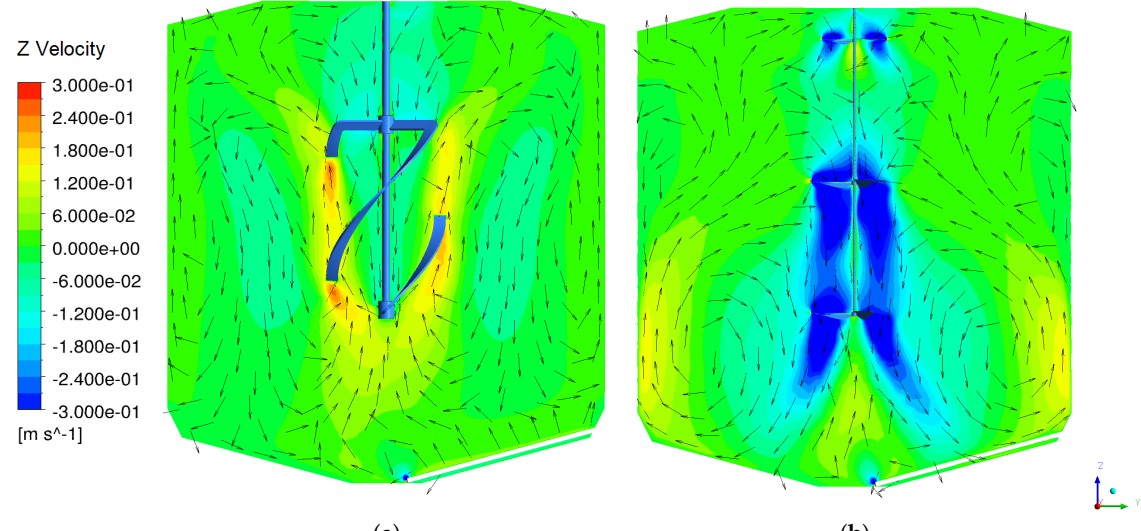

**Figure 8.** Contour and normalised vector plots of the z velocity for the helical (**a**) and SCABA (**b**) agitators at a TS concentration of 7.5%.

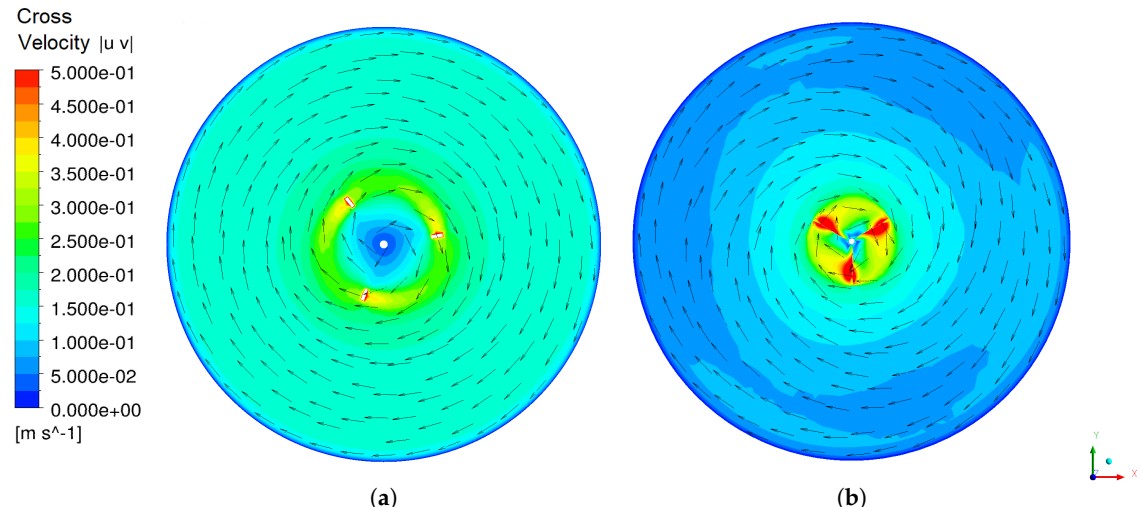

**Figure 9.** Contour and normalised vector plots of the cross velocities, $|\boldsymbol{u} + \boldsymbol{v}|$, for the helical (**a**) and SCABA (**b**) agitators at a height of 12.75 m from the bottom at a TS concentration of 7.5%.

### 5.5.3. Velocity Contributions

In this section, we will analyse the velocity contributions of the two agitators in the same reactor, as illustrated in Figure 10, in order to understand and assess the mixing capabilities of the generated flow fields of the two agitators at various TS concentrations. For the helical agitator, velocities $v \geq 0.1$ m/s contribute to 60–98% of the reactor volume across varying TS concentrations. In comparison, the same velocities only contribute between 12–67% of the reactor volume for the SCABA agitator and the best SCABA velocity contribution profile, TS 5.4%, is only slightly better than the worst case helical profile at TS 12.1%.

The SCABA agitator has the best mixing levels at TS concentrations 5.4% and 2.5%, which is ideal as the Rossau reactor operates between TS concentrations 2.4–4.3%. The velocity contribution profile with the fastest velocities is found at TS concentration 5.4% for the SCABA agitator, not at TS concentration 2.5%. The TS 5.4% fluid is approximately 3–4 times more viscous than the 2.5% fluid, so we expect faster dissipation of the small scale velocities and, therefore, larger velocity contributions for the slower velocities. However, increasing the viscosity also increases the rate of diffusion, such that a larger region of fluid near the rotating agitator is influenced by the rotation; more similar to rigid body rotation.

Overall, the observed faster hydrodynamic flow field in Figure 7 that is induced by the far field design of the helical device results in significantly better velocity contributions and, therefore, improved mixing potential in the reactor compared with the SCABA agitator. However, both agitator designs for all TS concentrations satisfy the minimum mixing level criteria that velocities $v \leq 0.01$ m/s must occupy less than 15% of the reactor volume [2], which implies that both designs would be appropriate for stirring the sludge in ADs for all TS concentrations. From these results, we can conclude that the generated flow field from the helical agitator produces an improved stirred reactor compared with the SCABA agitator. However, in order to properly assess the two designs we need to consider the power demand required to generate the flow field for each agitator; this is explored in the next section.

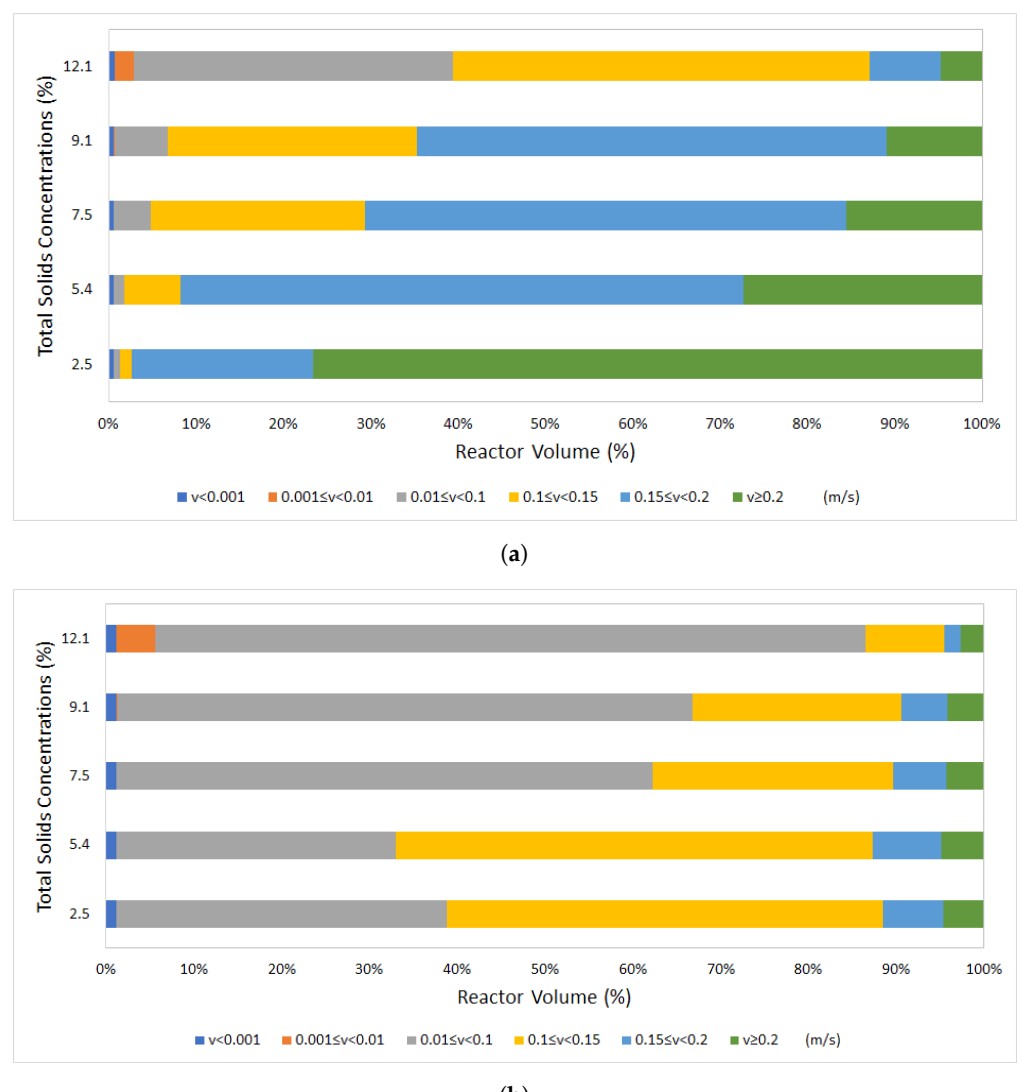

**Figure 10.** Velocity contributions for the helical (**a**) and SCABA (**b**) agitators at different TS concentrations.

### 5.5.4. Power Demand Calculation

In order to assess the operating efficiency of the two agitators, the power demand for both agitators is calculated for varying TS concentrations, as illustrated in Figure 11. The calculated power demand for the SCABA agitator at TS 2.5% and 5.4% is approximately 900 W. The working Rossau reactor with the SCABA agitator operates between TS concentrations 2.4–4.3% and has an average daily operation of 1500 W; if we account for the rotational direction changes during daily operation as described in Section 2.2, the calculated 900 W is a reasonable value for the power demand for this single rotational direction operation. At TS concentrations 2.5–9.1%, the SCABA agitator has approximately the same power demand for all four concentrations, which implies that the increase in viscosity across the four concentrations is not enough to impede on the local generated forces from the rotation of the SCABA design to increase the required energy to operate. In comparison, the helical agitator has a gradual increase in power demand as the viscosity increases from TS 2.5% to 9.1%. The larger surface area and slower rotational velocity of the helical device means the additional viscous forces from the increasing TS concentrations are significant enough to require additional energy to stir the more viscous TS concentrations. Both of the agitators have a significant increase in power demand for the TS 12.1% fluid implying the viscosity is large enough for this concentration to significantly resist the rotation of both agitators and, therefore, requires a substantial increase in power. When comparing

the two agitators, the helical agitator is more energy efficient across all TS concentrations, requiring 17–35% less energy than the SCABA agitator across the 12.1–2.5% TS concentrations, respectively. The helical agitator is both more energy efficient and produces a better mixing flow field than the SCABA agitator, which implies that the helical device is an improved design over the SCABA agitator for stirring slurries in AD.

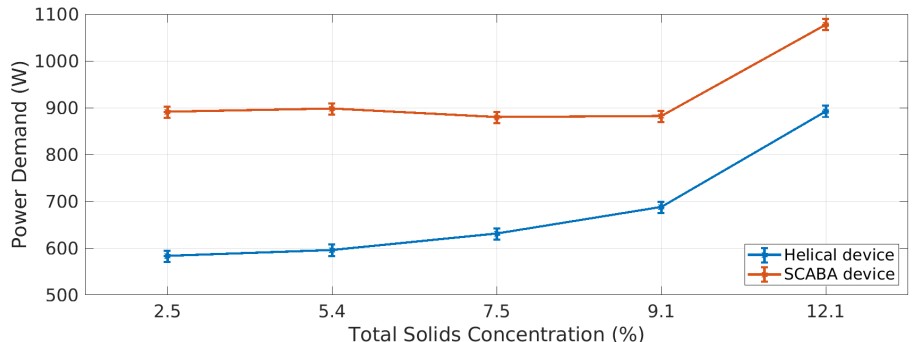

**Figure 11.** Power demand calculation for the helical and SCABA agitators at different TS concentrations.

*5.6. Improved Digester Operation: Power Saving Prospects*

Finally, the potential power saving prospects of using the helical agitator, which results in an improved digester operation, are discussed. Because the possible savings of electrical energy depend on the TS concentration, the focus of the discussion will be on TS concentrations of 2.5% and 7.5%. The reduced power demand of the Rossau wastewater treatment plant's anaerobic digester with the installation of the helical agitator, specifically 35% and 21% at a TS concentration of 2.5% and 7.5%, respectively, is used as the basis to estimate the potential power savings throughout the United Kingdom (UK) and Austria. The assumption made is that all treatment plants, except for very small sewage treatment plants, are technically upgraded similarly to the installation of the helical agitator in the Rossau digester. Small treatment plants are excluded because the possible savings per population equivalent (PE60, [35]) are expected to be much higher than for the Rossau digester. At the same time, many small treatment plants do not even have any anaerobic digester on-site and, if it exists, the operators may not be able to fund the acquisition costs that are required for the digester upgrade.

The potential savings of electrical energy throughout the UK and Austria are extrapolated from the reduced annual energy consumption of the Rossau digester. The saved energy from the two ADs is divided by the capacity of the Rossau plant (400.000 PE60), giving us a saved energy per PE. This calculation gives us a saved energy of 48.6 kJ/PE60 and 28.1 kJ/PE60 at a TS concentration of 2.5% and 7.5%, respectively. These values multiplied by the countries' capacity for sewage treatment, the total PE design capacity for the country excluding very small plants, produces the potential annual energy savings across the country. The UK's large treatment works have a total design capacity of 91 million PE60 [36] and, therefore, the estimated annual energy savings are calculated as 4.42 TJ and 2.55 TJ for a TS concentration of 2.5% and 7.5%, respectively. In Austria, with a total design capacity of 22 million PE60 [37], the estimated annual energy saving amounts to 1.07 TJ and 0.62 TJ for a TS concentration of 2.5% and 7.5% respectively. Since, apart from the initial digester upgrade, no additional effort is needed for the daily digester operation, the energy savings accumulate substantially over the years.

Besides that, the proposed digester upgrade with the installation of the helical agitator is also expected to enhance the biogas production as a result of the more homogeneous flow field with reduced velocity gradients. Furthermore, much higher medium-term savings are possible if anaerobic digesters are operated at higher TS concentrations, where the helical agitator still ensures the required level of mixing. Consequently, not only the electrical energy consumption, but the full operating costs as well as building and acquisition costs may be substantially reduced if smaller reactor volumes are sufficient.

## 6. Conclusions

In this paper, a CFD model has been developed in order to investigate the mixing capabilities and power consumption of two different agitator designs in the stirring of non-Newtonian slurries in anaerobic digestion. Lab-scale experiments with agitator designs have been conducted and used to validate the CFD methodology. The agitator devices were implemented into full-scale AD reactor simulations, in which the setup was based on the real working operating conditions of the Rossau wastewater treatment plant. A mesh refinement study has been completed and mesh independence has been confirmed for both mixing devices at full-scale and lab-scale.

Experimental investigations using a linear scaled reactor and ultrasonic flow field measurements were conducted in order to validate the CFD methodology. Water-glycerol mixtures were used to replicate the wastewater slurry at various TS concentrations. The average velocity values were calculated along 2D plane profiles at specific height increments and compared with equivalent CFD simulations results. The simulations matched very well for both fluids and agitator designs, illustrating the validity of the CFD modelling choices, such that these choices were adopted in the full-scale simulations.

A MRF study was conducted in order to investigate the impact of aspect ratio on the predicted hydrodynamic field and it was concluded that an aspect ratio size of 1.5–2.0 is recommended for modelling openly stirred AD reactors. The influence of the non-Newtonian behaviour has been analysed and found that only small regions near the agitator blades initiated the behaviour at TS concentrations of 7.5% and higher. However, it was found that even small regions of non-Newtonian behaviour could impact the results and, therefore, it is important to include it in the modelling of higher TS concentrated slurries.

A case study comparing the two agitators designs has been conducted in order to evaluate the two designs. Analysis of the velocity contributions and power demand showed that the helical device could achieve higher levels of mixing at lower power consumption than the SCABA agitator for all TS concentrations. This is due to the helical blades driving the flow at larger radii when compared with the SCABA device; the fluid driven by the helical device has a shorter distance to travel to reach the outer reactor regions and, therefore, can reach these region before the velocities and energy of the fluid has decayed away. This means the helical agitator can achieve the desired mixing levels at lower relative rotational velocities and power consumption than the SCABA device. The lower rotational velocities required for the helical device create lower shear rates than the SCABA agitator, reducing the influence of the non-Newtonian behaviour, which is a more desirable operating condition for the biological processes. Finally, the potential power savings prospects for the installation of the helical agitator at a TS concentration of 2.5% and 7.5% across the UK and Austria was assessed and found that a energy saving of: 4.42 TJ and 2.55 TJ for a TS concentration of 2.5% and 7.5%, respectively, for the UK; and 1.07 TJ and 0.62 TJ for a TS concentration of 2.5% and 7.5%, respectively, in Austria. From this case study, we have shown that the implementation of the novel helical agitator design into ADs can lead to more energy efficient operation.

**Author Contributions:** Conceptualization, A.O., M.M. and D.B.; methodology, A.O. and T.N.; software, A.O., T.N. and P.F.; validation, T.N. and A.O.; formal analysis, A.O., T.N. and M.M.; investigation, A.O. and T.N.; resources, A.O. and M.M.; data curation, A.O., P.F., T.N. and M.M.; writing—original draft preparation, A.O.; writing—review and editing, A.O., M.M., D.B., T.N. and M.C.-V.; visualization, A.O. and T.N.; supervision, M.M., D.B., M.C.-V. and A.S.; project administration, M.M., A.O. and D.B.; funding acquisition, M.M. All authors have read and agreed to the published version of the manuscript.

**Funding:** This research is partly funded by the Austrian Federal Ministry for Agriculture, Regions and Tourism in collaboration with the KPC and the Tyrolean Science Fund. AO was supported by EPSRC CDT in Fluid Dynamics EP/L01615X/1.

**Acknowledgments:** The simulation work was undertaken on ARC2 and ARC3, part of the High Performance Computing facilities at the University of Leeds, UK. The collaboration was possible thanks to the John Fox Award.

**Conflicts of Interest:** The authors declare no conflict of interest. The funders had no role in the design of the study; in the collection, analysis, or interpretation of data; in the writing of the manuscript, or in the decision to publish the results.

## Abbreviations

The following abbreviations are used in this manuscript:

CFD　　　Computational Fluid Dynamics
AD　　　 Anaerobic Digester
TS　　　　Total Solids
MRF　　　Multiple Reference Frame
FVM　　　Finite Volume Method
RANS　　 Reynolds-Averaged Navier-Stokes
PE60　　　Population Equivalent

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
