# Peer review of "Modelling Mechanically Induced Non-Newtonian Flows to Improve the Energy Efficiency of Anaerobic Digesters"

_water, doi:10.3390/w12112995_

Round 1

Reviewer 1 Report

The paper reports elaborate work examining designs for anaerobic digesters with respect to energy efficiency. Intensive work has been performed, experimentally and numerically that is presented in detail. The presentation is well organized and important information concerning the experiments and the modelling are outlined. The study is surely of interest not only for the audience of the WATER journal. Aside from the findings the report highlights the effectiveness of joint cooperation between experimentalists and modellers.

The only shortcoming of the manuscript is that some figures are not presented optimally. This concerns mainly the readability of figure text, if printed in the format of the manuscript. Here the details:

Figure 3: The axes labels are hardly, some legends completely unreadable.

Figures 4 and 5: As a basic orientation the legend text should be in a size comparable to the text in the body. There is space enough aside of the figures to improve in that respect.

Figure 6: The color plots are too small to see any details. I recommend to use only one color legend (as that is the same for both subplots) and to increase subfigure (a) to the same height as subfigure (b)

Figure 11: There is enough space to increase the fontsize to a size similar to the text font of the body.

Author Response

Reviewer 1 Changes Log

Comment

Changes

Figure 3: The axes labels are hardly, some legends completely unreadable.

The axes labelling and legend font size has been increased to make the plots easily readable.

Figures 4 and 5: As a basic orientation the legend text should be in a size comparable to the text in the body. There is space enough aside of the figures to improve in that respect.

The plot orientation has been flipped to reduce white space, font size increased to improve readability and overall figure structure has been modified to improve the illustration.

Figure 6: The color plots are too small to see any details. I recommend to use only one color legend (as that is the same for both subplots) and to increase subfigure (a) to the same height as subfigure (b)

The two plots have been combined with a single colour legend and the image sizes have been increased as well as trimmed so that only the informative parts of the plots remain.

Figure 11: There is enough space to increase the fontsize to a size similar to the text font of the body

The total font size has been increased to better match the text in the paper.

Reviewer 2 Report

This paper combines the experiments and CFD modelling to study the mixing of non-Newtonian flows and operating conditions of an anaerobic digester. The paper is well organized and suitable for publication in the journal. I have some suggetsions for the present version.

(1) In Fig.5, the measurement and simulations are not matched well near the bottom (height less than 0.07 m). Have you attempted to tune the bottom roughness? You may need to adjust bottom roughness to decrease the difference.

(2) Figs. 4 and 5 also show the significant variations of the measurements. You may need to detail the measurement method and provide the measurement uncertainty in the manuscript.

(3) Figure qualities are not good enough, and the fonts in some figures need to be enlarged.

Author Response

Reviewer 2 Changes Log

Comment

Changes

(1) In Fig.5, the measurement and simulations are not matched well near the bottom (height less than 0.07 m). Have you attempted to tune the bottom roughness? You may need to adjust bottom roughness to decrease the difference.

The simulations are using the enhanced wall function to resolve the near wall effects and we have refined the mesh near the walls. Hence, the y+ values are small enough such that we resolve the “laminar layer” near the walls. Additionally, this model does not require a wall roughness constant as the near wall features are resolved accurately. Further investigations showed that a change of the wall roughness constant does not significantly change the flow field near the base since the interaction of the fluid with the agitation devices prevail.

We ran additional simulations using the standard wall function that requires a wall roughness constant and analysed if this varied the results to confirm this. It was found that the wall roughness constant did not have an impact on the calculated values. I think the main reason for this is that the outer AD wall effects are not significant to the impact the generated bulk flow field. Additionally, the CFD data plotted is calculated from a weighted average of all the 2d planar velocities on a 2d slice through the reactor at a specific height. This means that most of the velocities considered for averaging are away from the walls and inside the bulk flow and, therefore, the effects of the walls on the flow are further supressed.   

(2) Figs. 4 and 5 also show the significant variations of the measurements. You may need to detail the measurement method and provide the measurement uncertainty in the manuscript

At the beginning of the experimental results and validation, there was a brief description of how each experimental data point was computed and how the error bars were calculated. This section has been extended further to explain the approach taken in detail.

(3) Figure qualities are not good enough, and the fonts in some figures need to be enlarged.

All figures (except for figure 1) have been modified to improve font size and minimise the white space.